# The elements of success in a comprehensive state-wide program to safely reduce the rate of preterm birth

John P. Newnham[1,2]*, Scott W. White[1,2], Han-Shin Lee[1], Catherine A. Arrese[2], Jared C. Watts[3,4], Michelle K. Pedretti[2,5], Jan E. Dickinson[1,2], Dorota A. Doherty[2]

1 Maternal Fetal Medicine Service, King Edward Memorial Hospital, Perth, Western Australia, Australia, 2 Division of Obstetrics and Gynaecology, The University of Western Australia, Perth, Western Australia, Australia, 3 Department of Obstetrics and Gynaecology, WA Country Health Service, Kimberley, Western Australia, Australia, 4 Rural Clinical School of Western Australia, The University of Western Australia, Broome, Western Australia, Australia, 5 Department of Ultrasound, King Edward Memorial Hospital, Perth, Western Australia, Australia

* john.newnham@uwa.edu.au

**Data Availability Statement:** The authors do not have permission to share patient-level data extracted from the Data Linkage Unit of the Department of Health of Western Australia. Data

## Abstract

### Background

In 2014, a whole-of-population and multi-faceted preterm birth prevention program was introduced in Western Australia with the single aim of safely lowering the rate of preterm birth. The program included new clinical guidelines, print and social media, and a dedicated new clinic. In the first full calendar year the rate of preterm birth fell by 7.6% and the reduction extended from the 28–31 week gestational age group upwards.

### Objective

The objective of this study was to evaluate outcomes in greater depth and to also include the first three years of the program.

### Study design

This was a prospective population-based cohort study of perinatal outcomes in singleton pregnancies before and after commencement of the program.

### Results

There was a significant reduction in preterm birth in the tertiary center which extended from 28 weeks gestation onwards and was ongoing. In non-tertiary centers there was an initial reduction, but this was not sustained past the first year. The greatest reduction was observed in pregnancies classified at first attendance as low risk. No benefit was observed in the private sector, but a significant reduction was seen in the remote region of the Kimberley where the program was first launched and vaginal progesterone had been made free-of-charge.

can only be made available to researchers who apply to the Department of Health of Western Australia's Human Research Ethics Committee (https://ww2.health.wa.gov.au/Articles/A_E/Department-of-Health-Human-Research-Ethics-Committee) and Data Linkage Unit (www.datalinkage-wa.org.au).

**Funding:** This study was supported by the National Health and Medical Research Council of Australia (Partnership Grant APP1151853 to JPN, DAD, www.nhmrc.gov.au), the Women and Infants Research Foundation of Western Australia (to JPN, DAD, www.wirf.com.au), Channel 7 Telethon (to JPN, DAD, www.telethon7.com) and private philanthropy. None of the study sponsors had any role in the design, data collection, analysis, interpretation of data, writing of the manuscript or decision to submit the paper for publication.

**Competing interests:** The authors have declared that no competing interests exist.

## Conclusion

Preterm birth rates can be safely reduced by a multi-faceted and whole-of-population program but the effectiveness requires continuing effort and will be greatest where the strategies are most targeted.

## Introduction

It is well recognized that preterm birth (PTB) is the single most important cause of perinatal mortality and morbidity with potential for life-long consequences [1–3]. Strategies to prevent this complication of pregnancy need to be given high priority.

In recent years there have been advances in our knowledge of how some pathways to early birth may be tackled [4–8]. Implementation of such strategies across populations, however, remains challenging due to the resources required, variations in access to health care across many communities, and the effectiveness of various interventions when applied into clinical practice.

In 2014, Western Australia hosted the introduction of a multi-faceted whole-of-population preterm birth prevention initiative based on seven interventions [9]. Only singleton pregnancies were targeted. The initiative involved print and social media (known as thewholeninemonths.com.au), an outreach program for health care professionals, and a new clinic in the tertiary center. The results during the first full calendar year (2015) were published and indicated a 7.6% reduction in preterm birth state-wide with a 20% reduction in the tertiary center [9]. The reductions in preterm birth extended from the 28–31 weeks group onwards.

The initial analysis involved only grouped data with no ability to dissect out important variables such as gestational age by individual weeks, the risk status of the pregnancy at first antenatal presentation, whether the birth followed spontaneous labour or medical intervention, public /private health insurance status or region within the state.

The purpose of this study was to analyze the effects of implementation of the program over the first three years (2015 to 2017 inclusive) and to determine those factors associated with success or failure of the program.

## Materials and methods

The study was approved by the Women and Newborn Health Service Human Research Ethics Committee (2016027EW) and the Health Department of Western Australia (EC00422).

Seven interventions were adopted for implementation state-wide, each chosen for the known evidence of effectiveness and suitability for utility in the Western Australian health care environment [9, 10] (Table 1).

The program was unofficially announced but fully outlined at a state-based conference of obstetricians at Broome in the Kimberley region in May 2014, and officially launched in Perth on November 17, 2014 (World Prematurity Day). Introduction was therefore step-wise over the six months of that year.

The program and its interventions were made known to the health care workforce and the general community by a combined print and social media campaign, accompanied by an outreach program. Overall, the program was known as thewholeninemonths[TM]. The print campaign included calls for action in local newspapers to direct people to the on-line resources together with regular production of magazines written specifically for the general public.

**Table 1. The key interventions included in the new clinical guidelines within the Western Australian preterm birth prevention initiative.**

1. Measurement of the length of the cervix to be included in all mid-pregnancy morphology scans, conducted routinely at 18–20 weeks' gestation. In those cases in which the cervix can be imaged clearly on transabdominal scan, a closed length from internal to external os of 35mm or more is adequate. In all other cases transvaginal scanning with an empty bladder is required at which a closed cervix length measured by this route of 25mm or less is considered shortened.

2. Natural vaginal progesterone 200mg tablet to be prescribed nightly for any case in which the cervix has been found on ultrasound imaging to be shortened between 16 and 24 weeks gestation. Treatment is to continue until 36 weeks gestation.

3. In cases in which the cervix length is <10mm on transvaginal imaging, management can include cervical cerclage, vaginal progesterone, or both.

4. Natural vaginal progesterone 200mg tablet to be prescribed nightly for any case in which there is a history of spontaneous preterm birth (with or without preterm prelabour rupture of membranes) between 20 and 34 weeks gestation and to be used each night from 16 to 36 weeks' gestation.

5. No pregnancy is to be ended prior to about 39 weeks' gestation unless there is a medical or obstetric indication.

6. Women who smoke should be identified and offered counselling through one of the well-established Quitline services offered through the Western Australian Department of Health.

7. A new dedicated and multi-disciplinary Preterm Birth Prevention Clinic established at the tertiary-level center for referral of high risk cases. Typically, a management plan is developed and the woman referred back to her referring physician when the high risk period is concluded. Maternal-fetal medicine specialists, ultrasound imaging facilities for cervix length measurement, mental health care and midwifery services are available at the clinic.

An outreach program was implemented to provide education for health care providers of all relevant disciplines near to their place of work. In general, this program involved 2-hour workshops in hospitals and health care centers throughout the state and included both didactic teaching and interactive learning techniques.

A new Preterm Birth Prevention Clinic was established at the tertiary perinatal center in November 2014. The purpose of this clinic and its associated resources was to accept referrals of cases at very high risk and to provide a focus for advice for health care practitioners.

Pregnancy data were obtained from the Midwives Notification System (MNS), under waiver of consent by the Ethics Committee, on all Western Australian births from 20 weeks gestation onwards in the years 2009 to 2017 using information recorded by the attending midwife.

Pregnancy information included maternal characteristics, medical and obstetric history, pregnancy complications, labor and births outcomes such as onset of labor, mode of birth, gestational age at birth and live born or stillborn status. History of PTB (present, absent or unknown) was derived from the consecutive births in years 2009 and 2017. All births that had occurred at the established tertiary center and the recently built evolving tertiary center were also identified.

The program and analysis involved only singleton pregnancies. Pregnancy terminations between 20 and 24 pregnancy weeks, identified by inductions resulting in intrapartum death at the established tertiary center, were excluded. PTB rate, overall and by gestational age (20–27, 28–31, 32–36 weeks) were examined. Comparisons were performed on births from the established tertiary center alone, the secondary and primary centers alone, and all births. Subgroup analyses were also performed on births classified in early pregnancy as being either at low or high risk of PTB. Births from the evolving tertiary center were only included in "all births", as the PTB rate in this center was substantially lower than at the established tertiary center, and substantially higher than at the non-tertiary centers.

Australia has universal public health care for all residents with the option of additional private health care. Data were analysed by public and private hospital of birth with both tertiary

level hospitals assumed to be entirely public and privately managed hospitals being assumed to be entirely private, although there would have been a few exceptions that could not be captured in the data set.

## Statistical methods

Binary logistic regression was used to model the probability of PTB and stillbirth. Nominal logistic regression was used to simultaneously model the probabilities of all gestational age-specific early births relative to term birth, because the assumption of proportional odds in the ordinal logistic regression was not satisfied. PTB rates in each year from 2009 to 2016 were compared to the PTB rate in 2017, with additional comparisons using year 2013 as a reference. The year effects on the rates of PTB were summarized using odds ratios (OR) and their 95% confidence intervals (shown in the supplementary tables). Maternal characteristics at the time of the first antenatal attendance including history of PTB were used to generate the probabilities of PTB and to then assign low or high risk. Derivation of the probability of preterm birth was performed on all pregnancies in years 2009–2017, hence was independent of the year when each pregnancy occurred. Logistic regressions to classify PTB risk were performed separately for nulliparous and parous women.

PTB rates over time were also investigated using run charts [11]. For this analysis, the baseline rates of PTB were calculated as the median bi-monthly rates, overall or gestational age-specific, from January 2013 to June 2014 before the unofficial partial introduction of the program. The assessment of changes in the PTB rates since June 2014 was conducted using the rules of probability with detection of runs (non-random patterns), shifts (six or more consecutive rates above/below the median), and trends (five or more consecutive rates increasing or decreasing).

SAS statistical software (proc genmod) (version 9.4, Cary, NC: SAS Institute Inc.) was used for data analysis. P-values<0.05 were considered statistically significant.

## Results

Across the study period there was a progressive increase in singleton births in Western Australia from 30,293 in 2009 to 33,492 in 2017. Table 2 shows the number of singleton births in this state in the years 2009 to 2017 for the state's established tertiary level center, the evolving tertiary level center, the two tertiary level centers combined, all others (secondary and primary)

Table 2. Number of singleton births in Western Australia between 2009 and 2017.

| Year | Established tertiary center* | Evolving tertiary center | Secondary/ primary centers | State overall |
|------|------|------|------|------|
| 2009 | 5413 | - | 24820 | 30233 |
| 2010 | 5510 | - | 24846 | 30356 |
| 2011 | 5405 | - | 25821 | 31226 |
| 2012 | 5663 | - | 27208 | 32871 |
| 2013 | 5452 | - | 27945 | 33397 |
| 2014 | 5476 | 114 | 28525 | 34115 |
| 2015 | 5319 | 2196 | 26429 | 33944 |
| 2016 | 5304 | 2727 | 26823 | 34854 |
| 2017 | 5455 | 2995 | 24987 | 33437 |
| All | 48997 | 8032 | 237404 | 294433 |

* 525 terminations performed between 20–24 pregnancy weeks at the established tertiary center were excluded (60, 59, 62, 55, 61, 59, 55, 59, and 55 in the respective years from 2009 to 2017)

and the state overall. The evolving tertiary level center commenced in 2014 with 114 births and increased to 2995 births in 2017.

The annual rates of PTB from singleton pregnancies are shown in Table 3 and the trends from 2013 to 2017 are shown in Fig 1. In the established tertiary level center, the rate of PTB decreased from 20.6% in 2013 to 16.2% in 2015 (P < 0.001 compared with 2013), remained significantly reduced from pre-intervention levels during 2016 and 2017 (both P< 0.001 compared with 2013) and remained similar during those two years (P = 0.795). For the emerging tertiary level center, the rate of PTB was 9.3% in 2015 and 8.2% in 2016 and 2017. In the secondary and primary centers combined, the PTB rate was 4.8% in 2013, 4.7% in 2015, 5.0% in 2016 and 5.3% in 2017 (P = 0.017 when 2017 is compared with 2013). For the state overall, the

**Table 3. Rates of preterm birth in singleton pregnancies stratified by gestational age and hospital level.**

| Year, GA wk | 20–27 | | 28–31 | | 32–36 | | <37 | |
|---|---|---|---|---|---|---|---|---|
| **Established tertiary*** | **n** | **%** | **n** | **%** | **n** | **%** | **n** | **%** |
| 2009 | 136 | 2.5 | 162 | 3.0 | 738 | 13.6 | 1036 | 19.1$^\uparrow$ |
| 2010 | 133 | 2.4 | 150 | 2.7 | 697 | 12.7 | 980 | 17.8 |
| 2011 | 122 | 2.3 | 139 | 2.6 | 713 | 13.2 | 974 | 18.0 |
| 2012 | 127 | 2.2 | 169 | 3.0$^\uparrow$ | 772 | 13.6 | 1068 | 18.9$^\uparrow$ |
| 2013 | 119 | 2.2 | 159 | 2.9$^\uparrow$ | 843 | 15.5$^\uparrow$ | 1121 | 20.6$^\uparrow$ |
| 2014 | 135 | 2.5 | 169 | 3.1$^\uparrow$ | 750 | 13.7 | 1054 | 19.3$^\uparrow$ |
| 2015 | 113 | 2.1 | 134 | 2.5 | 617 | 11.6 | 864 | 16.2 |
| 2016 | 134 | 2.5 | 157 | 3.0$^\uparrow$ | 642 | 12.1 | 933 | 17.6 |
| 2017 | 117 | 2.1 | 136 | 2.5 | 717 | 13.1 | 970 | 17.8 |
| **Secondary/primary centres** | **n** | **%** | **n** | **%** | **n** | **%** | **n** | **%** |
| 2009 | 38 | 0.2 | 37 | 0.2 | 945 | 3.8$^\downarrow$ | 1020 | 4.1$^\downarrow$ |
| 2010 | 36 | 0.1 | 23 | 0.1 | 1083 | 4.4$^\downarrow$ | 1142 | 4.6$^\downarrow$ |
| 2011 | 47 | 0.2 | 28 | 0.1 | 1091 | 4.2$^\downarrow$ | 1166 | 4.5$^\downarrow$ |
| 2012 | 47 | 0.2 | 28 | 0.1 | 1239 | 4.6$^\downarrow$ | 1314 | 4.8$^\downarrow$ |
| 2013 | 49 | 0.2 | 30 | 0.1 | 1251 | 4.5$^\downarrow$ | 1330 | 4.8$^\downarrow$ |
| 2014 | 45 | 0.2 | 26 | 0.1 | 1264 | 4.4$^\downarrow$ | 1335 | 4.7$^\downarrow$ |
| 2015 | 37 | 0.1 | 25 | 0.1 | 1184 | 4.5$^\downarrow$ | 1246 | 4.7$^\downarrow$ |
| 2016 | 44 | 0.2 | 43 | 0.2 | 1248 | 4.7 | 1335 | 5.0 |
| 2017 | 42 | 0.2 | 34 | 0.1 | 1242 | 5.0 | 1318 | 5.3 |
| **State overall** | **n** | **%** | **n** | **%** | **n** | **%** | **n** | **%** |
| 2009 | 174 | 0.6$^\uparrow$ | 199 | 0.7$^\uparrow$ | 1683 | 5.6$^\downarrow$ | 2056 | 6.8$^\downarrow$ |
| 2010 | 169 | 0.6$^\uparrow$ | 173 | 0.6 | 1780 | 5.9$^\downarrow$ | 2122 | 7.0 |
| 2011 | 169 | 0.5 | 167 | 0.5 | 1804 | 5.8$^\downarrow$ | 2140 | 6.9$^\downarrow$ |
| 2012 | 174 | 0.5 | 197 | 0.6 | 2011 | 6.1 | 2382 | 7.2 |
| 2013 | 168 | 0.5 | 189 | 0.6 | 2094 | 6.3 | 2451 | 7.3 |
| 2014 | 180 | 0.5 | 195 | 0.6 | 2016 | 5.9$^\downarrow$ | 2391 | 7.0$^\downarrow$ |
| 2015 | 156 | 0.5 | 168 | 0.5 | 1991 | 5.9$^\downarrow$ | 2315 | 6.8$^\downarrow$ |
| 2016 | 183 | 0.5 | 205 | 0.6 | 2104 | 6.0 | 2492 | 7.1 |
| 2017 | 162 | 0.5 | 192 | 0.6 | 2181 | 6.5 | 2535 | 7.6 |

Note that 8032 births (678 preterm births: 14 between 20–27 weeks, 38 between 28–31 weeks and 628 between 32–36 weeks) from the evolving tertiary center were excluded.

Statistical significance is shown by upwards arrows (↑, dark grey) for higher rates and by downwards arrows (↓, light grey) for lower rates relative to the 2017 rates.

Statistical adjustments were made for all maternal characteristics known at the time of the first antenatal visit used to derive low or risk of PTB.

Logistic regression analyses are shown in S1–S4 Tables.

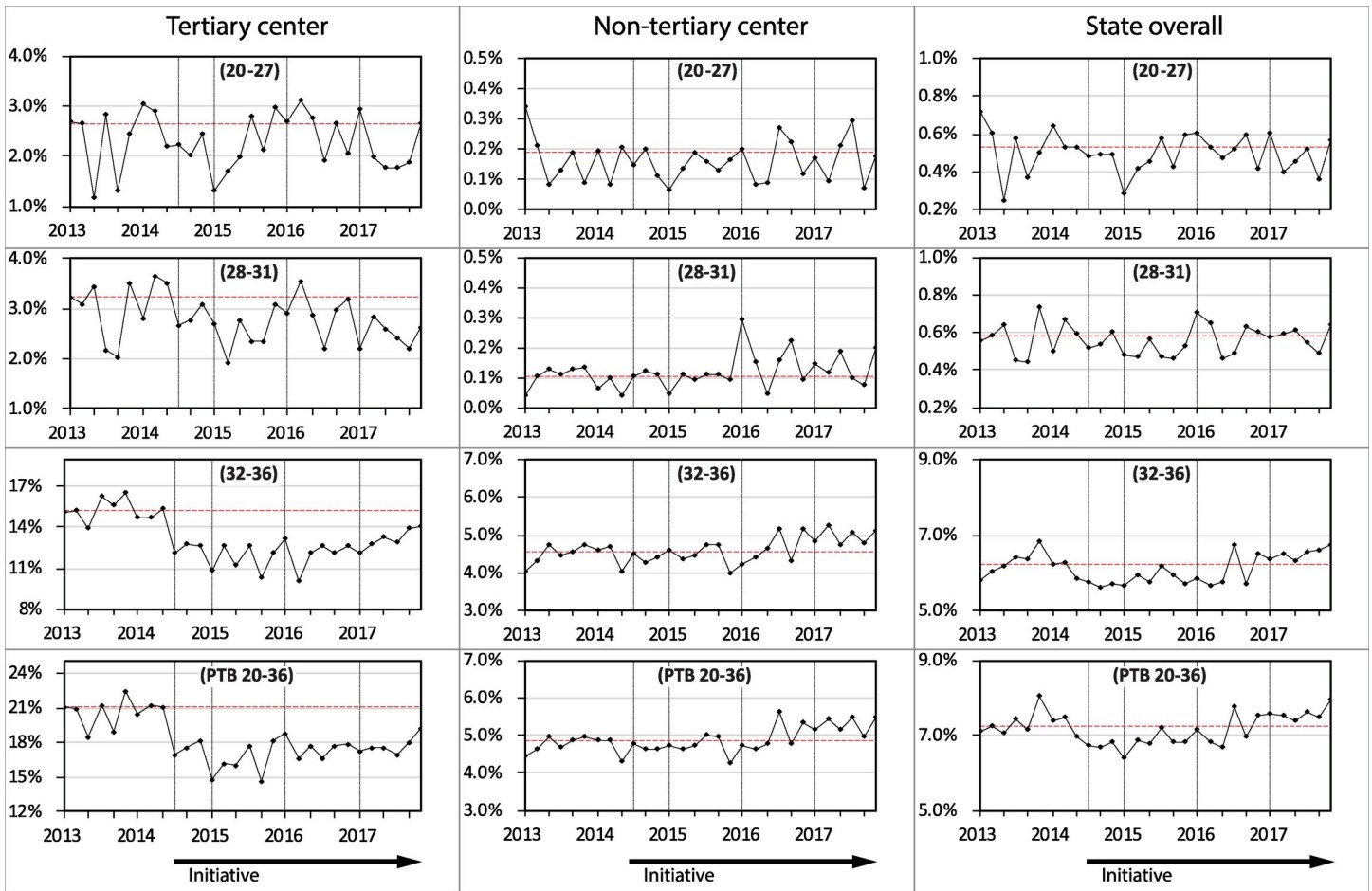

**Fig 1. Preterm birth rates between 2013 and 2017 in singleton pregnancies stratified by hospital level and the state of Western Australia overall (n = 169,747).** The gestational age groups in weeks are shown in brackets.

PTB rate decreased from 7.3% in 2013 to 7.0% in 2014, 6.8% in 2015, and then increased to 7.1% in 2016 and 7.6% in 2017.

In the established tertiary level center, the rate of PTB in the 32–36 week gestational age group fell from 15.5% in 2013 to 13.1% in 2017 (P<0.001) (Table 3). In the 28–31 week group, the rate in 2017 was 2.5%, which was also significantly lower than the rate of 2.9% in 2013 (P = 0.019). There were similar rates in the 20–27 week group across the years. In the secondary and primary centers combined, there was no difference in rates in the 32–36 week ages between 2013 and 2015 (P = 0.995) but there was an increase in 2017 when compared with 2013 (P = 0.014).

All pregnancies were classified as being at low or high risk of PTB based on maternal characteristics that were known at the time of the first antenatal visit (S5 Table). The odds ratios were 7.5 for prior PTB, 5.9 for pre-existing diabetes, 3.9 for previous stillbirth, 3.2 for pre-existing hypertension, 2.3 for indigenous ethnicity, 2.0 for grand-multiparity, 1.9 for smoking during pregnancy, 1.6 for maternal age <20 years, 1.2 for age > 34 years, 1.5 for IVF conception, and 1.2 for low socio-economic index. Tables 4 and 5 and Fig 2 (and S1 Fig) show the PTB rates for pregnancies classified using these risk factors. In the established tertiary level center the PTB rate declined following introduction of the Initiative in the pregnancies classified as

**Table 4. Rates of preterm birth in singleton pregnancies at low risk of preterm birth stratified by gestational age and hospital level.**

| Year, GA wk | 20–27 | | 28–31 | | 32–36 | | <37 | |
|---|---|---|---|---|---|---|---|---|
| **Established tertiary** | n | % | n | % | n | % | n | % |
| **2009** | 93 | 2.4 | 110 | 2.8[↑] | 440 | 11.3[↑] | 643 | 16.4[↑] |
| **2010** | 69 | 1.7 | 102 | 2.6[↑] | 422 | 10.6 | 593 | 14.9[↑] |
| **2011** | 70 | 1.8 | 74 | 2.0 | 414 | 10.9[↑] | 558 | 14.7[↑] |
| **2012** | 83 | 2.1 | 89 | 2.2[↑] | 410 | 10.3 | 582 | 14.5[↑] |
| **2013** | 71 | 1.8 | 81 | 2.1[↑] | 474 | 12.3[↑] | 626 | 16.1[↑] |
| **2014** | 86 | 2.2 | 94 | 2.4[↑] | 411 | 10.5[↑] | 591 | 15.2[↑] |
| **2015** | 80 | 2.1 | 69 | 1.8 | 311 | 8.1 | 460 | 12.0 |
| **2016** | 78 | 2.1 | 85 | 2.3[↑] | 322 | 8.9 | 485 | 13.1 |
| **2017** | 64 | 1.8 | 55 | 1.6 | 331 | 9.3 | 450 | 12.8 |
| **Secondary/primary centres** | n | % | n | % | n | % | n | % |
| **2009** | 22 | 0.1 | 24 | 0.1 | 683 | 3.3[↓] | 729 | 3.5[↓] |
| **2010** | 20 | 0.1 | 15 | 0.1 | 805 | 3.9 | 840 | 4.0 |
| **2011** | 27 | 0.1 | 20 | 0.1 | 808 | 3.8 | 855 | 4.0 |
| **2012** | 28 | 0.1 | 13 | 0.06[↓] | 874 | 3.9 | 915 | 4.0 |
| **2013** | 32 | 0.1 | 19 | 0.1 | 882 | 3.8 | 933 | 4.0 |
| **2014** | 30 | 0.1 | 14 | 0.06[↓] | 864 | 3.7 | 908 | 3.9 |
| **2015** | 28 | 0.1 | 15 | 0.1 | 812 | 3.7 | 855 | 3.9 |
| **2016** | 27 | 0.1 | 24 | 0.1 | 827 | 3.7 | 878 | 4.0 |
| **2017** | 26 | 0.1 | 25 | 0.12 | 785 | 3.9 | 836 | 4.1 |
| **State overall** | n | % | n | % | n | % | n | % |
| **2009** | 115 | 0.5[↑] | 134 | 0.6[↑] | 1123 | 4.6 | 1372 | 5.6 |
| **2010** | 89 | 0.4 | 117 | 0.5[↑] | 1227 | 4.9 | 1433 | 5.8[↑] |
| **2011** | 97 | 0.4 | 94 | 0.4 | 1222 | 4.8 | 1413 | 5.6 |
| **2012** | 111 | 0.4 | 102 | 0.4 | 1284 | 4.8 | 1497 | 5.6 |
| **2013** | 103 | 0.4 | 100 | 0.4 | 1356 | 5.0 | 1559 | 5.7 |
| **2014** | 116 | 0.4 | 108 | 0.4 | 1276 | 4.6 | 1500 | 5.4 |
| **2015** | 112 | 0.4 | 89 | 0.3 | 1256 | 4.5 | 1457 | 5.3 |
| **2016** | 108 | 0.4 | 111 | 0.4 | 1296 | 4.6 | 1515 | 5.4 |
| **2017** | 93 | 0.4 | 94 | 0.4 | 1252 | 4.8 | 1439 | 5.5 |

Note that 6620 births (448 preterm births: 10 between 20–27 weeks, 21 between 28–31 weeks and 417 between 32–36 weeks) from the evolving tertiary center were excluded.

Statistical significance is shown by upwards arrows (↑, dark grey) for higher rates and by downwards arrows (↓, light grey) for lower rates relative to the 2017 rates.

Statistical adjustments were made for all maternal characteristics known at the time of the first antenatal visit used to derive low or risk of PTB.

Logistic regression analyses are shown in S1–S4 Tables.

low risk from 16.1% in 2013 to 12% in 2015 (P<0.001) and to 12.8% in 2017 (P<0.001). For high risk pregnancies (Table 5) there also was a significant reduction from 2013 (31.5%) to 2017 (26.9%) (P = 0.018) although the relative reduction of 15% in high risk pregnancies was less than the reduction of 21% in the many more low risk pregnancies. In the non-tertiary centers the increase in rate of PTB occurred primarily in the pregnancies classified as high risk (in 2013 the rate was 8.5% and in 2017 was 10.3%) whereas in the low risk cases the rates remained similar (in 2013 the rate was 4% and in 2017 was 4.1%).

During the three year period of study, the proportion of pregnancies classified as high risk increased significantly within the established tertiary level center (from 29% in 2014 to 35.5% in 2017, P <0.001); in the secondary and primary centers combined (from 17.3% in 2014 to

**Table 5. Rates of preterm birth in singleton pregnancies at high risk of preterm birth stratified by gestational age and hospital level.**

| Year, GA wk | 20–27 | | 28–31 | | 32–36 | | <37 | |
|---|---|---|---|---|---|---|---|---|
| **Established tertiary** | n | % | n | % | n | % | n | % |
| 2009 | 43 | 2.9 | 52 | 3.5 | 298 | 20.1 | 393 | 26.5 |
| 2010 | 64 | 4.2$^\uparrow$ | 48 | 3.1 | 275 | 18.0$^\downarrow$ | 387 | 25.3 |
| 2011 | 52 | 3.3 | 65 | 4.1 | 299 | 18.7 | 416 | 26.0 |
| 2012 | 44 | 2.7 | 80 | 4.8 | 362 | 21.8 | 486 | 29.2 |
| 2013 | 48 | 3.1 | 78 | 5.0 | 369 | 23.5$^\uparrow$ | 495 | 31.5$^\uparrow$ |
| 2014 | 49 | 3.1 | 75 | 4.7 | 339 | 21.4 | 463 | 29.2 |
| 2015 | 33 | 2.2 | 65 | 4.4 | 306 | 20.6 | 404 | 27.2 |
| 2016 | 56 | 3.5 | 72 | 4.5 | 320 | 20.1 | 448 | 28.1 |
| 2017 | 53 | 2.7 | 81 | 4.2 | 386 | 20.0 | 520 | 26.9 |
| **Secondary/primary centres** | n | % | n | % | n | % | n | % |
| 2009 | 16 | 0.4 | 13 | 0.3 | 262 | 6.3$^\downarrow$ | 291 | 7.0$^\downarrow$ |
| 2010 | 16 | 0.4 | 8 | 0.2 | 278 | 7.0$^\downarrow$ | 302 | 7.6$^\downarrow$ |
| 2011 | 20 | 0.5 | 8 | 0.2 | 283 | 6.6$^\downarrow$ | 311 | 7.3$^\downarrow$ |
| 2012 | 19 | 0.4 | 15 | 0.3 | 365 | 8.0$^\downarrow$ | 399 | 8.7$^\downarrow$ |
| 2013 | 17 | 0.4 | 11 | 0.2 | 369 | 7.9$^\downarrow$ | 397 | 8.5$^\downarrow$ |
| 2014 | 15 | 0.3 | 12 | 0.2 | 400 | 8.1$^\downarrow$ | 427 | 8.6$^\downarrow$ |
| 2015 | 9 | 0.2 | 10 | 0.2 | 372 | 8.4$^\downarrow$ | 391 | 8.8$^\downarrow$ |
| 2016 | 17 | 0.4 | 19 | 0.4 | 421 | 9.1 | 457 | 9.8 |
| 2017 | 16 | 0.3 | 9 | 0.2 | 457 | 9.7 | 482 | 10.3 |
| **State overall** | n | % | n | % | n | % | n | % |
| 2009 | 59 | 1.0 | 65 | 1.2 | 560 | 9.9$^\downarrow$ | 684 | 12.1$^\downarrow$ |
| 2010 | 80 | 1.5$^\uparrow$ | 56 | 1.0 | 553 | 10.0$^\downarrow$ | 689 | 12.5$^\downarrow$ |
| 2011 | 72 | 1.2 | 73 | 1.2 | 582 | 9.9$^\downarrow$ | 727 | 12.4$^\downarrow$ |
| 2012 | 63 | 1.0 | 95 | 1.5 | 727 | 11.7$^\downarrow$ | 885 | 14.2$^\downarrow$ |
| 2013 | 65 | 1.0 | 89 | 1.4 | 738 | 11.8$^\downarrow$ | 892 | 14.2$^\downarrow$ |
| 2014 | 64 | 1.0 | 87 | 1.3 | 740 | 11.3$^\downarrow$ | 891 | 13.6$^\downarrow$ |
| 2015 | 44 | 0.7 | 79 | 1.3 | 735 | 11.7$^\downarrow$ | 858 | 13.7$^\downarrow$ |
| 2016 | 75 | 1.1 | 94 | 1.4 | 808 | 12.0$^\downarrow$ | 977 | 14.5 |
| 2017 | 69 | 1.0 | 98 | 1.4 | 929 | 12.9 | 1096 | 15.3 |

Note that 1412 births (230 preterm births: 4 between 20–27 weeks, 15 between 28–31 weeks and 211 between 32–36 weeks) from the evolving tertiary center were excluded.

Statistical significance is shown by upwards arrows (↑, dark grey) for higher rates and by downwards arrows (↓, light grey) for lower rates relative to the 2017 rates.

Statistical adjustments were made for all maternal characteristics known at the time of the first antenatal visit used to derive low or risk of PTB.

Logistic regression analyses in pregnancies at low risk are shown in S6 and S8–S10 Tables, and in pregnancies at high risk in S7 and S11–S13 Tables.

18.8% in 2017, P <0.001) and in the state overall (19.2% in 2014 to 21.5% in 2017, P <0.001). The increased percentage translates to 770 additional high risk cases in 2017 compared to 2014. Of these additional cases, 355 would have been at the tertiary centre and 375 in the non-tertiary centres.

In the tertiary level center the reduction in PTB was in cases both of medically initiated birth and those resulting from spontaneous labor (Figs 3 and 4). In contrast, in the non-tertiary centers the rate of medically initiated PTB increased. Preterm births following spontaneous labor decreased in the tertiary center, largely within the 32 to 36 week age group.

The dedicated PTB Prevention Clinic commenced in November 2014. Until the end of 2017, a total of 435 women had attended the clinic and 370 pregnancies (93 in 2015, 120 in

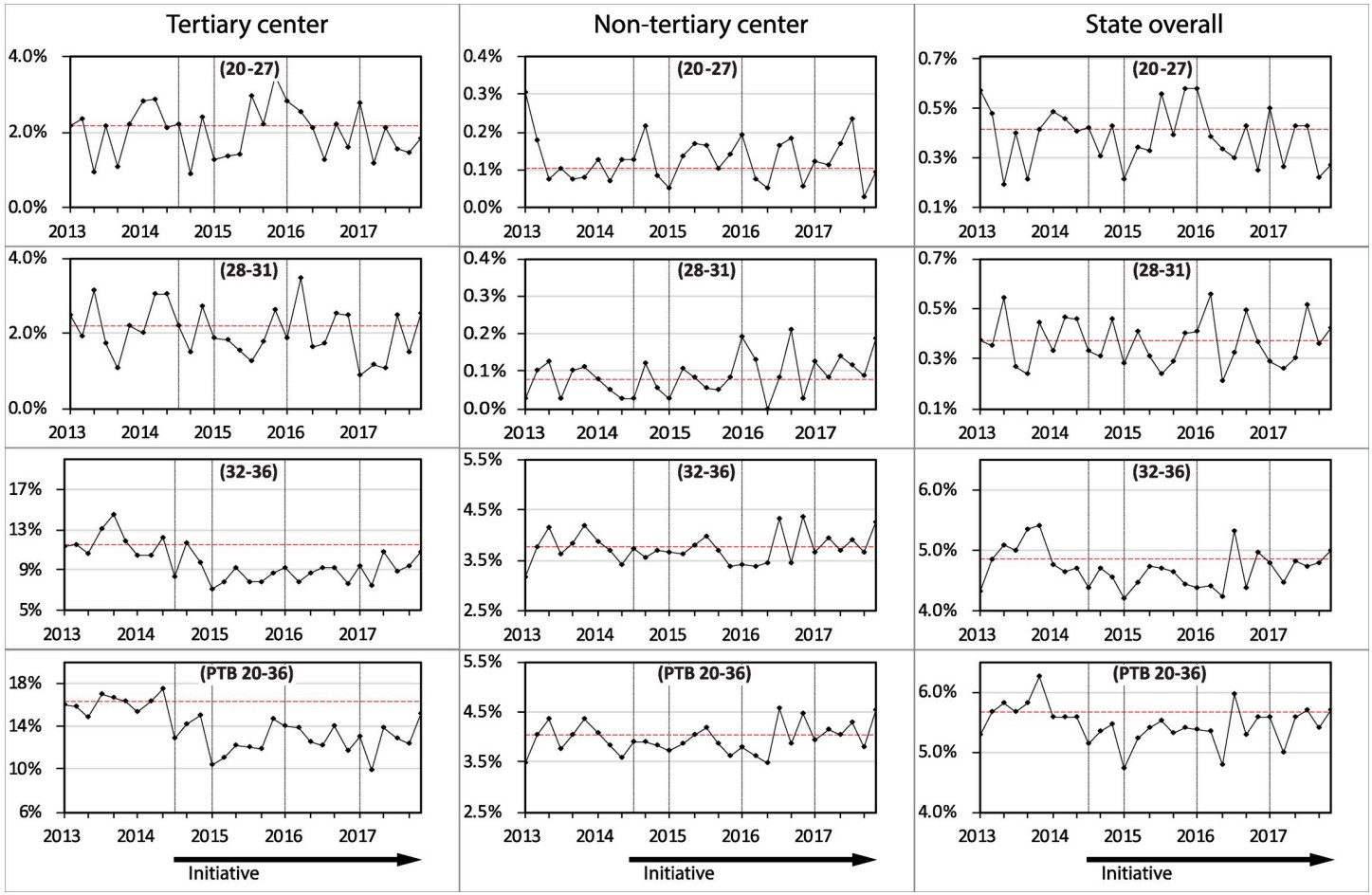

**Fig 2. Preterm birth rates between 2013 and 2017 in singleton pregnancies classified at first antenatal visit as low risk stratified by hospital level and in the state of Western Australia overall.** The gestational age groups in weeks are shown in brackets. (n = 136,756).

2016 and 157 in 2017) were concluded. The median number of visits was 3 (range, 1–8) in 2015 and increased to 4 (range 1, 13) in 2016–2017. The median gestational age at the first visit overall was 13.1 weeks (range, 6.6–27.0) with median gestational age of 13.6 (range 9.3–26.3) in 2015, 12.9 (range 6.6–25.4) in 2016 and 13.3 (range 6.9–27.0) in 2017. Women with histories of early PTBs (69.7%, n = 258 of 370), recurrent pregnancy losses (22.2%, n = 82 of 370), autoimmune conditions (7.6%, n = 28 of 370), uterine anomalies (8.4%, n = 31 of 370), placental risk factors (8.4%, n = 31 of 370), and/or a history of cone biopsies or other ablative procedures of the cervix (17.3%, n = 64 of 370) were referred. One hundred and eighty three women (49.5%) were treated with vaginal progesterone and 98 women (26.5%) by cervical cerclage, 123 women (33.2%) had medical interventions, and 97 women (26.2%) required mental health intervention. Excluding 14 pregnancy losses (8, 3, and 3 in the consecutive years), 244 of 356 women delivered at term (68.5%) (63.4%, 66.7% and 66.9% in the consecutive years). Two hundred ninety out of 356 women (81.5%) referred to the dedicated Clinic remained in the obstetric care at the tertiary centre until delivery (71.0%, 82.5%, and 79.6% in the consecutive years from 2015 to 2017).

In this state, the tertiary level center is almost exclusively public accouting for about 18% of the state's total births, with 38% in the non-tertiary public hospitals and 42% in private

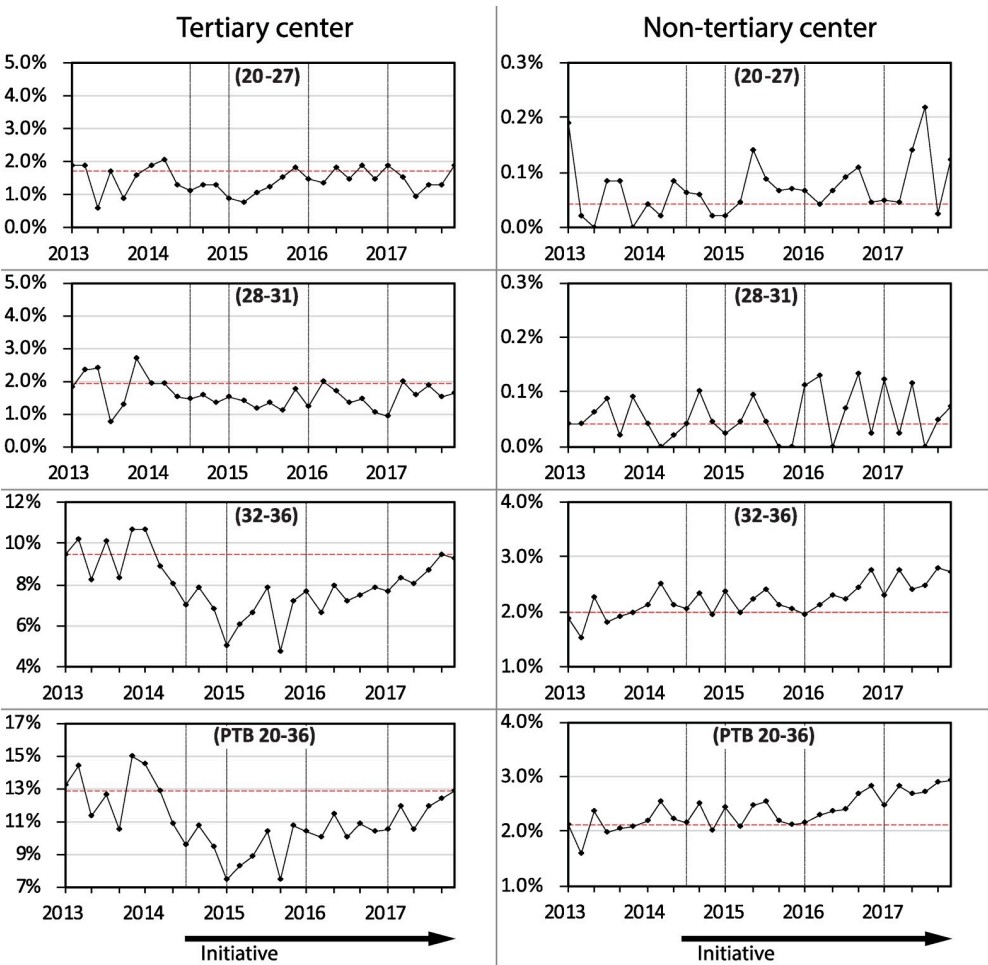

**Fig 3. Medically initiated preterm birth rates between 2013 and 2017 in singleton pregnancies stratified by hospital level.** The gestational age groups in weeks are shown in brackets. (n = 88,381).

hospitals. PTB rates by public and private hospital of birth are shown in Fig 5. Overall, the PTB rate in private hospitals was significantly higher (OR = 1.41; 95% CI 1.36 to 1.47, P <0.001) than in non-tertiary public hospitals and there was a non-significant increase in PTB rates in both sectors over the study period. The reduction in PTB rates in the state overall over the study period was largely within the tertiary center, with a lesser reduction in 2015 in the non-tertiary public centers, and no reduction in the private centers.

Outside the capital city of Perth, Western Australia is divided into seven regions. The most northerly region, and with one of the highest rates of PTB and indigenous pregnancies in Australia, is the Kimberley. There was a significant reduction in PTB rates in women with a Kimberley place of residence and classified as low risk from 6.5% in 2013 to 2.2% in 2017 (P<0.001) (Fig 6). No such reduction was observed in Kimberley pregnancies classified as high risk nor in any of the other rural health care regions.

Stillbirth rates from the year 2009 to 2017 are shown in Table 6. There were no significant changes in the rates of stillbirth at the established tertiary center, the non-tertiary centers or the state overall following introduction of the Initiative or in the years that followed.

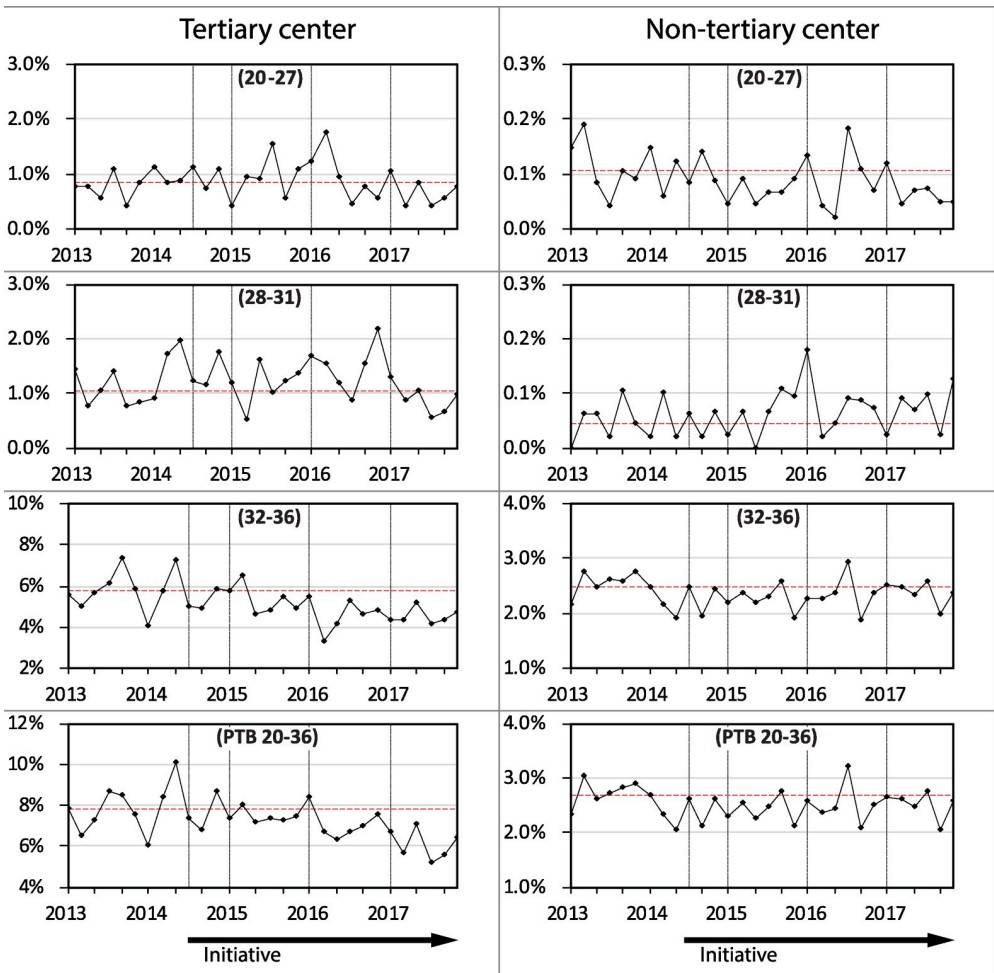

**Fig 4. Preterm birth rates following spontaneous labor between 2013 and 2017 in singleton pregnancies stratified by hospital level.** The gestational age groups in weeks are shown in brackets. (n = 81,366).

## Comment

### 1. Principal findings

The results of this study have shown that the whole-of-population whole-of-state preterm birth prevention program introduced into Western Australia during 2014[9] resulted in an on-going reduction over the following years in singleton preterm births in the established tertiary level perinatal center but initial success in the non-tertiary sector was followed by a rise in rates. Within the tertiary center the reduction in early births included both the early preterm gestational ages and the later preterm periods. There also was a reduction in PTB in pregnancies classified at first antenatal visit as low risk in women resident in the Kimberley region, which is a region in Australia which previously had one of the highest rates of PTB.

### 2. Results

The rapid decrease in PTB rates following introduction of the program in 2014 most likely resulted from changed behaviour of health care practitioners and pregnant women aiming to avoid unnecessary early birth [12, 13]. There was also a significant decrease in births at earlier

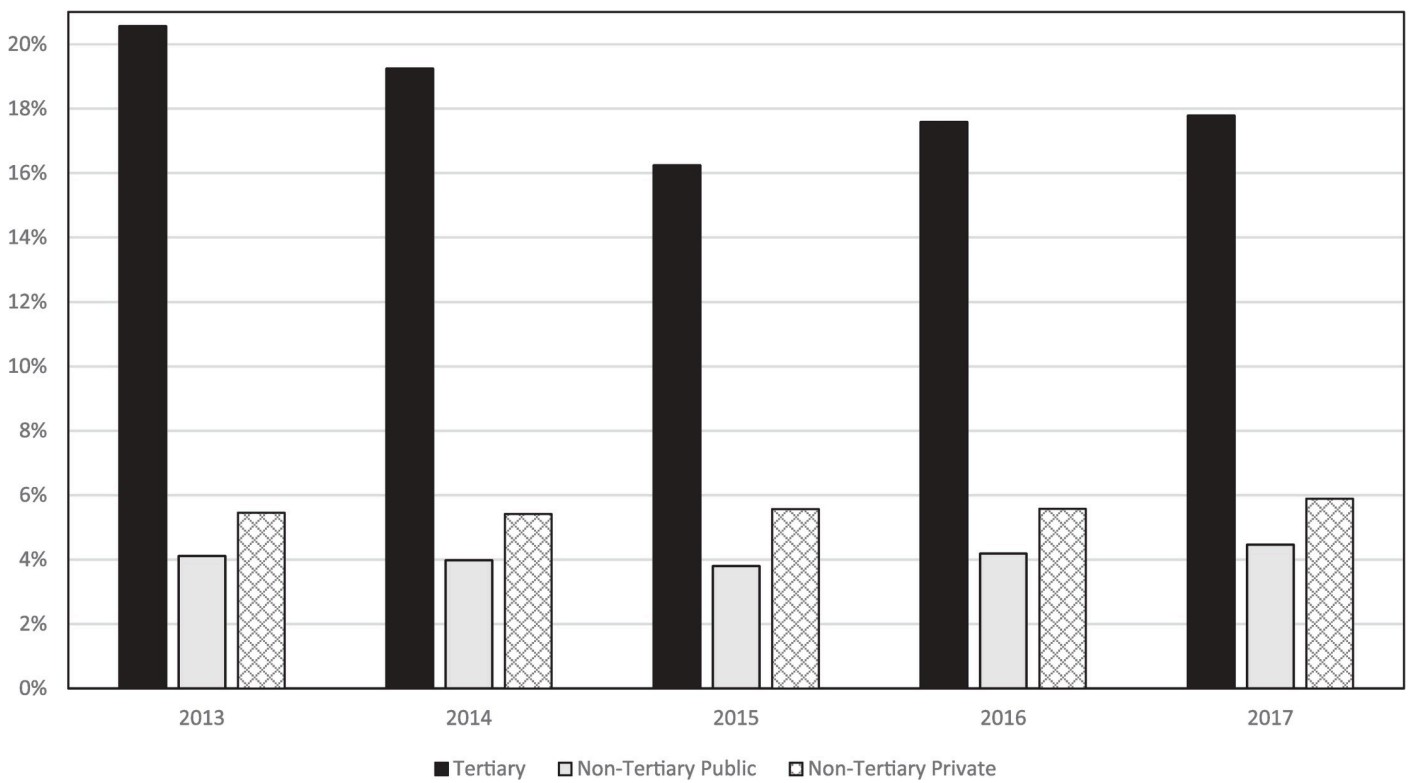

**Fig 5. Preterm birth rates between 2013 and 2017 in singleton pregnancies in the tertiary hospital (black bars), non-tertiary public hospitals (gray bars) and non-tertiary private hospitals (hatched bars).** (n = 161,715).

gestational ages and inspection of the run charts indicates the effect was delayed when compared with the effects at later ages. It is most likely the reduction at earlier ages resulted from

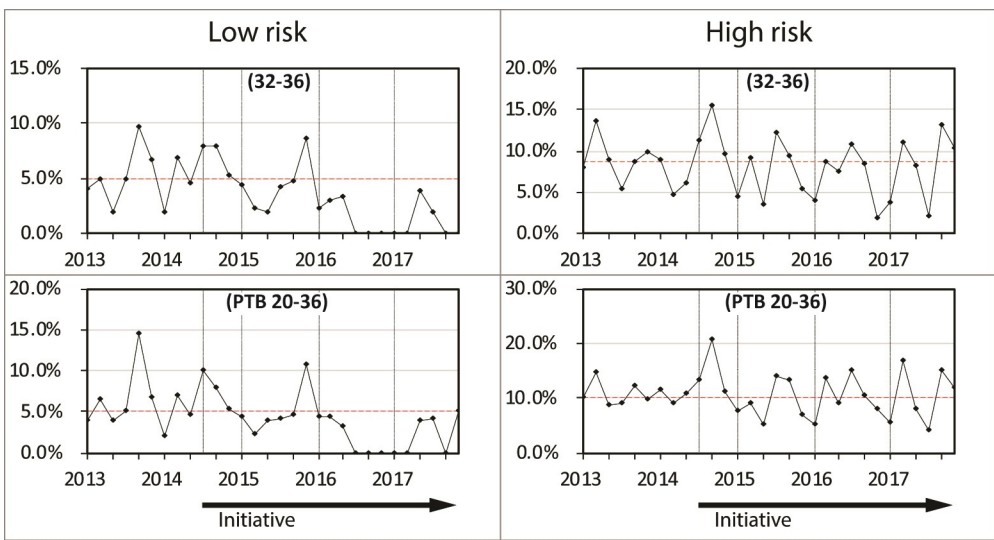

**Fig 6. Preterm birth rates between 2013 and 2017 in singleton pregnancies in women with the Kimberley region as their place of residence stratified by level of risk at the time of their first antenatal visit.** The gestational age groups in weeks are shown in brackets. (low risk n = 1,420 and high risk, n = 1,838).

**Table 6. Stillbirths in singleton pregnancies between 2009 and 2017.**

| Year | N | | | | Rate per 1000 | | | |
|------|---------------------------|------------------------|----------------------------|------------------|---------------------------|------------------------|----------------------------|------------------|
|      | Established tertiary center | Evolving tertiary center | Secondary/ primary centers | State overall | Established tertiary center | Evolving tertiary center | Secondary/ primary centers | State overall |
| 2009 | 80  | -  | 73  | 153  | 14.8 | -   | 2.9  | 5.1  |
| 2010 | 67  | -  | 74  | 141  | 12.2 | -   | 3.0  | 4.6  |
| 2011 | 75  | -  | 108 | 183  | 13.9 | -   | 4.2* | 5.9* |
| 2012 | 82  | -  | 78  | 160  | 14.5 | -   | 2.9  | 4.9  |
| 2013 | 64  | -  | 63  | 127  | 11.7 | -   | 2.3  | 3.8  |
| 2014 | 74  | 0  | 83  | 157  | 13.5 | 0.0 | 2.9  | 4.6  |
| 2015 | 71  | 11 | 69  | 151  | 13.3 | 5.0 | 2.6  | 4.4  |
| 2016 | 57  | 6  | 87  | 150  | 10.7 | 2.2 | 3.2  | 4.3  |
| 2017 | 70  | 10 | 78  | 158  | 12.8 | 3.3 | 3.1  | 4.7  |
| All  | 640 | 27 | 713 | 1380 | 13.1 | 3.4 | 3.0  | 4.7  |

Note that 525 pregnancy terminations between 20–24 weeks excluded (60, 59, 62, 55, 61, 59, 55, 59 and 55 in years 2009–2017 respectively).

Stillbirth rates are compared using logistic regression analysis with year 2017 as a reference, univariately and after adjustment for maternal age$\geq$35, nulliparity, grand-multiparty, ethnicity, smoking during pregnancy, maternal asthma, low socioeconomic status, history of stillbirth, placental abruption, antepartum haemorrhage for reasons other than placental abruptions and placenta praevia, gestational diabetes and 'other' pregnancy complications. Significantly higher rates, both univariately and with adjustments for maternal characteristics are marked with an asterisk.

Logistic regression analyses are shown in S14 Table.

mid-pregnancy screening of cervical length and progesterone therapy because unnecessary iatrogenic preterm birth would not occur at such early ages.

In the secondary and primary sectors the initial improvements in PTB rates were not sustained over the three year period. At least in part, this effect may have resulted from the increasing risk profile of the obstetric population over the three years. However, the changing face of the prevention program may also have contributed. At commencement of the state-wide program great efforts were made by all means possible to provide information to health care practitioners and women on the various interventions. After the first year, the print and social media campaigns continued but the face-to-face travelling outreach program was not continued. It would appear likely that the discordant outcomes in the tertiary versus non-tertiary environments resulted from effective daily interactions between members of the preterm birth prevention service and other health care personnel in the tertiary centre, while the lack of on-going presence in the other centres made it difficult to sustain the benefits. Any program based heavily on education of practitioners and the general public will require ongoing maintenance of a high level of interaction. Indeed, within the world of pregnancy care, the patient population is constantly changing and any educational program will need to maintain a high level of intensity.

All pregnancies were classified based on risk factors that would have been evident at the time of first presentation for antenatal care. Within the tertiary center the greatest reduction in PTB rates was in the pregnancies classified as low risk. There was also a reduction in the pregnancies classified as high risk, but the effect was less. It would appear that the greatest benefit in applying this bundle of interventions is to be found in those pregnancies thought initially to not be at increased risk. Identification of a shortened cervix in asymptomatic women in mid-pregnancy, and avoidance of unnecessary induction or cesarean section, would both seem to target such otherwise low risk pregnancies. In contrast, the increase in PTB rates in non-tertiary centers in the latter years occurred in pregnancies classified as high risk. It is likely that the practitioners working in such centers responded to risk factors by initiating early births in

the hope of preventing stillbirth, whereas those in the tertiary centers may have been surrounded by monitoring and resources that provided confidence in the safety to prolong the pregnancy.

A significant reduction was observed in PTB rates in women resident in the Kimberley region and classified as low risk. This region is known to have one of the nation's highest rate of preterm birth and low birthweight [14] and 58% of births are to indigenous women in whom the PTB rate is typically 14% or more [15, 16]. A strategic decision was made at the outset to specifically target this region. The program was first launched in the major center in the Kimberley called Broome in May 2014; progesterone was made free-of-charge across the region where it is relatively expensive elsewhere in the state; and the outreach program visited all significant towns. High quality ultrasound services are provided to pregnant women, even in the most remote communities, through the Royal Flying Doctor Service with additional funding and support from the national "closing the gap" program. The resulting success suggests that even in such an environment of high medical risk, the rate of PTB can be reduced with education combined with a complete subsidy of the cost of progesterone.

## 3. Clinical implications

The finding that the multi-faceted program has had its greatest benefit in pregnant women classified as low risk at the time of their first antenatal attendance suggests that preterm birth prevention programs centred purely on establishment of a dedicated clinic for referral of high risk cases may be missing their greatest opportunity to benefit the population as a whole. Including low risk as well as high risk cases in a program is far more challenging than providing care within the confines of a referral-based hospital environment, but the obvious benefits for women and their pregnancies should justify the additional effort and expenditure involved in making preventative strategies available to all.

Australia has a universal public health care system available to all residents. Private insurance is optional and enables health care in a private hospital. Within Western Australia, there is no private hospital providing tertiary level care and pregnancies requiring elevation to this level of care are in general referred to the tertiary center and thereafter classified as public. While the rate of PTB in the private sector overall was higher than in the non-tertiary public sector, both sectors experienced a similar but non-significant increase in PTB during the study period. It would now appear that future success in these sectors will require more intensive outreach education and perhaps policy change.

One of the most important outcomes to monitor during introduction of a preterm birth prevention program is the rate of stillbirth [12]. Delaying the time of birth by necessity carries risk of fetal death but the risk can be mitigated by clinical judgement and tests of fetal wellbeing. There were no significant changes to the rate of stillbirth across the state during or following introduction of this program confirming safety of the package of interventions.

## 4. Research implications

A major unanswered question is the potential effect of introduction of midwifery continuity of care in various parts of the state across the years of this study. Midwife-led continuity of care has been defined as care where the midwife is the lead professional in the planning, organisation and delivery of care given to a woman from initial booking to the postnatal period [17]. Within the established tertiary center the proportion of pregnancies managed by midwife-led care increased from 10 to 18% during the three year period. Meta-analysis of randomized controlled trials comparing midwifery-led care with standard management has shown a significant 24% reduction in the rate of preterm birth [17]. It remains unknown if this model of care

is as effective when introduced into the clinical environment in Western Australia, and if it is as effective, the mechanism by which it is beneficial [18]. Further research is required to determine if implementation is as efficacious as has been suggested by the trials, and how it may interact with a co-existing preterm birth prevention program. Since the midwives on the program were also promoting the package of interventions it is entirely possible that the two initiatives were synergistic rather than competitive.

## 5. Strengths and limitations

The strength of this project has been its whole-of-population capacity and multi-faceted suite of interventions, coupled with extensive data collection and analysis. In this regard it had great power to discover any effects at a population level, but unlike randomised controlled trials of single interventions, is limited by its ability to dissect out the causal pathways leading to its outcomes.

It remains unknown if the reduction in PTB rates in the tertiary sector may have in part resulted from changing referral patterns but the increasing PTB risk profile within the hospital population would provide confirmation that the prevention strategies themselves were being effective. To the best of our knowledge, this study is the first multi-faceted whole-of-population and whole-of-geographic region program to be published, precluding comparison with similar work in different healthcare environments.

## 6. Conclusions

A whole-of-population multi-faceted preterm birth prevention program based on existing knowledge has the capacity to safely lower the rate of early birth. The greatest effectiveness is observed in those centers where the prevention strategies are most targeted and appears to focus preferentially on pregnancies that would have been considered initially as low-risk. Any benefit, however, is likely to dissipate in regions not exposed to continuing education. Further advances will require expanded educational programs, possibly the free prescription of vaginal progesterone, policy change and new discoveries.

## Supporting information

**S1 Table. Risk of preterm birth stratified by hospital level in unadjusted and adjusted models.**
(PDF)

**S2 Table. Gestational age specific risk of preterm birth in the established tertiary level center in unadjusted and adjusted models.**
(PDF)

**S3 Table. Gestational age specific risk of preterm birth in the secondary/primary level centres in unadjusted and adjusted models.**
(PDF)

**S4 Table. Gestational age specific risk of preterm birth overall in unadjusted and adjusted models.**
(PDF)

**S5 Table. Unadjusted risk of preterm birth for maternal characteristics known at the time of the first antenatal visit in singleton pregnancies in years 2009–207.**
(PDF)

**S6 Table. Risk of preterm birth in low risk singleton pregnancies stratified by hospital level in unadjusted and adjusted models.**
(PDF)

**S7 Table. Risk of preterm birth in high risk singleton pregnancies stratified by hospital level in unadjusted and adjusted models.**
(PDF)

**S8 Table. Gestational age specific risk of preterm birth in low risk singleton pregnancies at tertiary level center in unadjusted and adjusted models.**
(PDF)

**S9 Table. Gestational age specific risk of preterm birth in low risk singleton pregnancies in secondary/primary level centers, in unadjusted and adjusted models.**
(PDF)

**S10 Table. Gestational age specific risk of preterm birth in low risk singleton pregnancies state-wide in unadjusted and adjusted models.**
(PDF)

**S11 Table. Gestational age specific risk of preterm birth in high risk singleton pregnancies at tertiary level center in unadjusted and adjusted models.**
(PDF)

**S12 Table. Gestational age specific risk of preterm birth in high risk singleton pregnancies in secondary/primary level centers, in unadjusted and adjusted models.**
(PDF)

**S13 Table. Gestational age specific risk of preterm birth in high risk singleton pregnancies state-wide in unadjusted and adjusted models.**
(PDF)

**S14 Table. Stillbirth by year table adjusted for maternal risk factors and pregnancy complications associated with stillbirth.**
(PDF)

**S1 Fig. Preterm birth rates between 2013 and 2017 in high risk pregnancies stratified by hospital level and the state overall.**
(PDF)

## Acknowledgments

The authors wish to thank Maureen Hutchinson, the Data Custodian for the Midwives Notification System for provision of the pregnancy data; and the WA Data Linkage Branch, in particular Mikhalina Dombrovskaya and Emma Douglas for their assistance.

The authors also wish to acknowledge the substantial contributions of Cate Belcher and Suzie Allen who were the lead midwives who contributed to the PTB Prevention Clinic and the outreach program.

We are grateful to the people who provided advice and enabled the Initiative to be endorsed by the Western Australian Department of Health (Professor Bryant Stokes, Professor Gary Geelhoed, Professor Tarun Weeramanthri, Dr Anne Karczub, Dr Janet Hornbuckle, Mr Graeme Boardley and Dr Dianne Mohen), The Western Australian branch of the Australian Medical Association (Dr Michael Gannon), and the Royal Australian and New Zealand College of Obstetricians and Gynaecologists (Professor Michael Permezel and Dr Vijay Roach).

Finally, we would like to thank the women and families of Western Australia who have responded so positively to this statewide initiative.

## Author Contributions

**Conceptualization:** John P. Newnham, Catherine A. Arrese, Dorota A. Doherty.

**Data curation:** John P. Newnham, Han-Shin Lee, Michelle K. Pedretti, Jan E. Dickinson, Dorota A. Doherty.

**Formal analysis:** John P. Newnham, Dorota A. Doherty.

**Funding acquisition:** John P. Newnham, Catherine A. Arrese, Dorota A. Doherty.

**Investigation:** John P. Newnham, Scott W. White, Han-Shin Lee, Jared C. Watts, Michelle K. Pedretti, Jan E. Dickinson.

**Methodology:** John P. Newnham, Dorota A. Doherty.

**Project administration:** John P. Newnham, Scott W. White, Catherine A. Arrese.

**Validation:** Jared C. Watts.

**Writing – original draft:** John P. Newnham, Dorota A. Doherty.

**Writing – review & editing:** John P. Newnham, Scott W. White, Han-Shin Lee, Catherine A. Arrese, Jared C. Watts, Michelle K. Pedretti, Jan E. Dickinson, Dorota A. Doherty.

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
