## [Decision Letter · Decision Letter 0]

4 May 2020

PONE-D-20-09304

The elements of success in a comprehensive state-wide program to safely reduce the rate of preterm birth

PLOS ONE

Dear Prof. Newnham,

Thank you for submitting your manuscript to PLOS ONE. After careful consideration, we feel that it has merit but does not fully meet PLOS ONE’s publication criteria as it currently stands. Therefore, we invite you to submit a revised version of the manuscript that addresses the points raised during the review process.

Both reviewers provided high praise for the conduct and reporting of this study. Further clarification or modifications in how the data were presented are suggested. 

We would appreciate receiving your revised manuscript by Jun 18 2020 11:59PM. To enhance the reproducibility of your results, we recommend that if applicable you deposit your laboratory protocols in protocols.io, where a protocol can be assigned its own identifier (DOI) such that it can be cited independently in the future. For instructions see: http://journals.plos.org/plosone/s/submission-guidelines#loc-laboratory-protocols

We look forward to receiving your revised manuscript.

Kind regards,

Christine E East

Academic Editor

PLOS ONE

Journal Requirements:

2. Please provide additional details regarding participant consent. In the ethics statement in the Methods and online submission information, please ensure that you have specified (a) whether consent was informed and (b) what type you obtained (for instance, written or verbal, and if verbal, how it was documented and witnessed). If the need for consent was waived by the ethics committee, please include this information.

Reviewers' comments:

Reviewer's Responses to Questions

**Comments to the Author**

1. Is the manuscript technically sound, and do the data support the conclusions?

Reviewer #1: Yes

Reviewer #2: Yes

2. Has the statistical analysis been performed appropriately and rigorously? 

Reviewer #1: Yes

Reviewer #2: Yes

3. Have the authors made all data underlying the findings in their manuscript fully available?

Reviewer #1: Yes

Reviewer #2: Yes

4. Is the manuscript presented in an intelligible fashion and written in standard English?

Reviewer #1: Yes

Reviewer #2: Yes

5. Review Comments to the Author

Reviewer #1: Globally rates of preterm birth continue to rise and as preterm birth remains a leading cause of neonatal morbidity and mortality, successful preterm birth prevention programs are significant and of interest to those working in maternity and perinatal healthcare including policy makers. The Western Australia state-wide preterm birth prevention program has been reported on previously but this manuscript contains new data including outcomes over a three year period since introduction (previously reported first year data only) and has attempted to identify which areas of the program are effective which has not been published previously.

The study has been undertaken with appropriate ethics approvals. The manuscript is well written and presented and includes appropriate statistical tests and analyses that are well described. Tables 3-5 are a little difficult to follow and although would require more space and table numbers, the inclusion of odds ratios in the supplementary tables were easier to understand and gave good context on size of effect. Study data availability is said to be fully available without restriction but only includes data within the manuscript and supporting files. It may be expected that additional data from the study exists but not used within the report.

The introduction and comment sections do not refer to other published literature within this field (with the exception of midwifery led models of care which were not part of the care bundle). The manuscript would be strengthened if results were given some context from other studies (or lack of them).

My main questions regarding this report relate to conclusions drawn regarding which elements of the program were effective and lack of data to support these. The manuscript in part acknowledges the challenges of identifying which elements of the package were successful but some additional data may help the reader to understand this. I have provided some examples below on this.

Through most of the report all preterm birth data is grouped together rather than by spontaneous and indicated/iatrogenic (or medically initiated) preterm birth. For example, it is suggested that the reduction in preterm birth at early gestational ages is due to detection of a short cervix in ‘low risk’ women who are subsequently the treated with vaginal progesterone and hence this would result in fewer spontaneous preterm births, and it is suggested that the reduction in preterm birth at later gestational ages is due to less indicated/iatrogenic births. Providing data by spontaneous and indicated/iatrogenic preterm birth would support these conclusions. Data by spontaneous and medically initiated is shown within the run charts in figs 3 and 4 but not used through results in the text or table.

Further support for the conclusion of a reduction in early preterm births (presumably spontaneous) being due to USS detection of short cervix and progesterone use could be provided by data on proportion of women undergoing an assessment of cervical length at mid-trimester scan, number with short cervix and number that received progesterone (if these data are available?).

Maternal characteristics to identify women at high and low risk of preterm birth include some risk factors for both spontaneous and medically initiated preterm birth, however, the majority of interventions within the package (exception avoiding birth <39 weeks) are focused on spontaneous preterm birth again provision of data by indication for preterm birth would be helpful.

A main conclusion is that there was a sustained reduction in preterm births for those women initially identified as low risk (and the reduction in risk was due to identification of a short cervix and use of progesterone). If this is true this is a highly significant finding as this represents the majority of the general population and hence opportunity for major impact. However, my interpretation of these data is that this was only applicable to those cared for in the established tertiary centre at a time when the number of women regarded as high risk increased. It seems such a significant shift in risk profile over only three years is higher than may be expected and it may be possible that greater awareness/reporting of risk status may have led those at higher risk of preterm birth being assessed at first visit as high risk hence overall reducing rates in an even more low risk group (adjOR in table S6 adjusts for risk factors, however, should all these women not have these risk factors). Further consideration of this should be discussed.

The reduction in rates of preterm birth in a disadvantaged group within the study is of particular interest (the Kimberly region) as being able to address higher rates of preterm birth for these women has significant implications for other disadvantaged groups. It has been suggested by the authors that free supply of progesterone was a factor in this reduction, however, as the group where the reduction was seen were considered low risk at first visit, it would suggest that these women were able to complete cervical length scanning (and progesterone was then prescribed as a consequence of a short cervix). Providing routine cervical length scanning in such a remote and disadvantaged area seems challenging and therefore some comment on how it was achieved would be helpful to the readership.

In summary I think this report provides valuable information that may be of use for those wishing to develop sate or even nationwide based preterm birth prevention programmes, however, inclusion of spontaneous and medically initiated preterm birth as a single group makes interpretation of data more difficult.

Reviewer #2: A study of an extraordinary program to reduce preterm birth on a statewide scale involving a number of elements and a co-ordinated effort.

The data from this study is extraordinarily valuable and should be able to provide direction as to whether our efforts are likely to succeed or not and also possibly which areas are most worthy of attention.

There is a wealth of detail in the reported numbers and also the analysis that has been performed however I was left with a few questions.

1. Did referrals to teh tertiary specialist preterm birth clinic increase over the study period or did they stay static? Do the authors have any idea of how many high risk women across the state were actually referred to the clinic? Was this different to previous referral patterns ie were women previously referred to the tertiary centre just not to the single clinic? Tertiary centre preterm birth rates remained lower and would seem to be one of the successes of the program but what happened to the women referred to the clinic? Were they sent back to their local centres or did they remain at the tertiary centre?

2. It was mentioned in the analysis that number of "high risk" women fluctuated from year to year but I couldn't work out whether this had been accounted for in the analysis at any point. Obviously preterm birth rates are very dependent on the risk profile of the population.

3. I found the numerical table data difficult to follow. Why was it presented from 2009 when the intervention started in late 2013? I found the extra numbers made it unnecessarily more complex. Perhaps a five year average could be included for the preceding years rather than the actual details of the years themselves. I also found the arrows in different types confusing. Could this information be included in the graphical representations which were much more easily intelligible?

6. PLOS authors have the option to publish the peer review history of their article (what does this mean?). If published, this will include your full peer review and any attached files.

Reviewer #1: No

Reviewer #2: No

---

## [Author Response · Author response to Decision Letter 0]

15 May 2020

Dear Dr East,

We are writing in reply to your email of Tuesday 5th May 2020 and attach our revised manuscript, with and without the changes being shown in track changes as requested.

First, we would like to thank the reviewers for their “high praise for the conduct and reporting of our study” and you for considering our manuscript further.

We will address the issues in turn.

Reviewer #1

• Consider inclusion of odds ratios in tables 3 and 4, as shown in the supplementary tables. We have chosen to present the data in these two tables as rates and statistical significance as this is the format we believe most meaningful for clinicians who may be considering adoption of a similar program to that described in this paper. Odds ratios are available in the supplementary tables for those interested in that form of presentation. Inclusion of odds ratios in the three tables would make them very large and difficult to read. We request permission to leave them unchanged but would happily add the odds ratios if requested.

• Consider adding reference to other similar studies in the literature in the introduction or discussion. We had not referred to other similar studies because, to the best of our knowledge, this is the first report of a multi-faceted whole of population and whole of geographic region program to be published. We have now added a new sentence to the Comment section, line 354: “To the best of our knowledge, this study is the first multi-faceted whole-of-population and whole-of-geographic region program to be published, precluding comparison with similar work in different healthcare environments.” 

• “Data by spontaneous and medically initiated is shown within the run charts in figs 3 and 4 but not used through results in the text or table”. We wish to draw your attention to lines 209 – 212: “In the tertiary level center the reduction in PTB was in cases both of medically initiated birth and those resulting from spontaneous labour (Figs 3 and 4). In contrast, in the non-tertiary centers the rate of medically initiated PTB increased. Preterm births following spontaneous labor decreased in the tertiary center, largely within the 32 to 36 week age group.” We have incorporated these observations in the conclusions we have drawn throughout the paper on our view that the reductions in early PTB most likely result from ultrasound screening of cervical length and progesterone medication, and reductions in late PTB most likely result from decreased medical intervention. We cannot provide definitive proof, however, without subjecting these concepts to a prospective RCT which for us would not be feasible or possible. 

• Can you provide data on cervix length findings and prescription of progesterone? The data for these issues are not available to us at this time and are the subject of current studies. We can assure you, however, based on our extensive role in the obstetric services in this state, that measurement of the length of the cervix at all mid-pregnancy scans is the standard of care and no exception has been seen by us since 2015. Quantification of vaginal progesterone usage is complex but in progress.

• …provision of data by indication for preterm birth would be helpful. We have included extensive data on the two categories of spontaneous and medically-initiated PTB in the various tables and figs in the manuscript and supplementary tables. Dissecting the clinical indications into finer categories, such as induction for preeclampsia and post-dates would be beyond our data set and we suggest add confusion to this complex clinical scenario. 

• Has greater awareness/reporting of risk status led those at higher risk of PTB being assessed at first visit as high risk hence overall reducing rates in an even more low risk group. This question by the reviewer is of great interest but we suggest does not apply in this study as the assessment of risk was not done by clinicians at the time but by us in retrospect as a description of maternal demographics and features. Our analysis was performed on all pregnancies between 2009 and 2017. The risk of preterm birth was assigned retrospectively using maternal characteristics recorded for all years and was independent of the year of pregnancy, thereby enabling us to assess any change in the level of preterm birth risk in the obstetric population. We have now added a sentence in the Statistical methods section, line 154 “Derivation of the probability of preterm birth was performed on all pregnancies in years 2009-2017, hence was independent of the year when each pregnancy occurred.” 

We also edited the supplementary table S5 caption by adding the years “2009-2017” at the end of the caption to clarify this point.

• The provision of clinical services in the remote region of the Kimberley is of great interest politically in our nation. Providing the highest possible level of care for people living in remote regions, and in particular indigenous people living in isolated communities is of very high priority. Extensive funding is provided through a national program called “closing the gap”. High quality ultrasound imaging services are provided even in the most remote regions via the Royal Flying Doctor Service and vaginal progesterone is provided free-of-charge as described in the manuscript. We have now added a sentence line 298: “High quality ultrasound services are provided to pregnant women, even in the most remote communities, through the Royal Flying Doctor Service with additional funding and support from the national “closing the gap” program. 

Reviewer #2

• Did referrals to the tertiary specialist preterm birth clinic increase over the study period or did they stay static? Do the authors have any idea of how many high risk women across the state were actually referred to the clinic? Was this different to previous referral patterns ie were women previously referred to the tertiary centre just not to the single clinic? Tertiary centre preterm birth rates remained lower and would seem to be one of the successes of the program but what happened to the women referred to the clinic? Were they sent back to their local centres or did they remain at the tertiary centre? The dedicated PTB Prevention Clinic commenced in mid-November 2014. Women at high risk of preterm birth were referred to the Clinic for consultation with the Maternal-Fetal Medicine specialists. Once the high risk period was over, they were referred back to their original physician. However, the majority of women (82%) who attended the Clinic remained at high risk and concluded their pregnancy at the tertiary level centre. Although the women referred to the PTB Prevention Clinic represented a small proportion of high risk women who accessed tertiary obstetric care, the lower rates of PTB achieved in the Clinic population were indicative of its success. 

We have added a paragraph line 213 to clarify this point: “The dedicated PTB Prevention Clinic commenced in November 2014. Until the end of 2017, a total of 435 women had attended the clinic and 370 pregnancies (93 in 2015,120 in 2016 and 157 in 2017) were concluded. The median number of visits was 3 (range, 1-8) in 2015 and increased to 4 (range 1, 13) in 2016-2017. The median gestational age at the first visit overall was 13.1 weeks (range, 6.6-27.0) with median gestational age of 13.6 (range 9.3-26.3) in 2015, 12.9 (range 6.6-25.4) in 2016 and 13.3 (range 6.9-27.0) in 2017. Women with histories of early PTBs (69.7%, n=258 of 370), recurrent pregnancy losses (22.2%, n=82 of 370), autoimmune conditions (7.6%, n=28 of 370), uterine anomalies (8.4%, n=31 of 370), placental risk factors (8.4%, n=31 of 370), and/or a history of cone biopsies or other ablative procedures of the cervix (17.3%, n=64 of 370) were referred. One hundred and eighty three women (49.5%) were treated with vaginal progesterone and 98 women (26.5%) by cervical cerclage, 123 women (33.2%) had medical interventions, and 97 women (26.2%) required mental health intervention. Excluding 14 pregnancy losses (8, 3, and 3 in the consecutive years), 244 of 356 women delivered at term (68.5%) (63.4%, 66.7% and 66.9% in the consecutive years). Two hundred ninety out of 356 women (81.5%) referred to the dedicated Clinic remained in the obstetric care at the tertiary centre until delivery (71.0%, 82.5%, and 79.6% in the consecutive years from 2015 to 2017).”

• It was mentioned in the analysis that number of "high risk" women fluctuated from year to year but I couldn't work out whether this had been accounted for in the analysis at any point. Obviously preterm birth rates are very dependent on the risk profile of the population. The number of high risk women fluctuated from year to year and our multivariable analyses were adjusted for all maternal characteristics known at the time of the first antenatal visit, which were used to derive low or high risk for preterm birth. We have now modified the footnotes in Tables 3, 4 and 5 as follows: “Statistical significance is shown by upwards arrows (↑) for higher rates and by downwards arrows (↓) for lower rates, relative to the 2017 rates. Statistical adjustments were made for all maternal characteristics known at the time of the first antenatal visit used to derive low or risk of PTB.”

We have also added two covariates, namely asthma and PTB history, to the lists of adjustments in the supplementary tables S1-4, S6-13. We apologise for this omission in the initial manuscript, which may have generated some confusion for the reviewer.

• Presentation of data from 2009 onwards. We have chosen to show the actual data from 2009 onwards to provide readers with the best possible chance of observing trends. Presentation as a five-year average would obscure potential findings. We suggest this extra step is of value but would re-present it as a five year average if requested.

• The arrows in different directions are confusing. We are keen to provide readers with as much information as possible in this very important table but have removed the arrows for univariate analyses to simplify tables 3-5 and reduce any potential for confusion. 

We hope we have addressed the questions from the reviewers adequately and would be pleased to address any issues that remain.

As the study is a retrospective analysis of patient data, consent was waived by the Ethics Committee. We added a comment, line 121 to provide this information.

We would like to rectify our misunderstanding concerning data availability. When stating that data were available without restriction, we were referring to our manuscript data but we are not authorised to share patient-level data, which are in the custody of the Department of Health of Western Australia and its Data Linkage Unit. Access to patient-level data is only granted to researchers who have applied to the Department of Health of Western Australia’s Human Research Ethics Committee (ww2.health.wa.gov.au/Articles/A_E/Department-of-Health-Human-Research-Ethics-Committee) and Data Linkage Unit (www.datalinkage-wa.org.au). We have added a paragraph line 41 to clarify this point.

Once again, we would like to thank you for considering our manuscript for publication in Plos one.

Kind regards

John (on behalf of all authors)

John P Newnham AM

2020 Senior Australian of the Year

---

## [Editor Report · Decision Letter 1]

19 May 2020

The elements of success in a comprehensive state-wide program to safely reduce the rate of preterm birth

PONE-D-20-09304R1

Dear Dr. Newnham,

We are pleased to inform you that your manuscript has been judged scientifically suitable for publication and will be formally accepted for publication once it complies with all outstanding technical requirements.

With kind regards,

Christine E East

Academic Editor

PLOS ONE
---

## [Editor Report · Acceptance letter]

26 May 2020

PONE-D-20-09304R1 

The elements of success in a comprehensive state-wide program to safely reduce the rate of preterm birth 

Dear Dr. Newnham:

I am pleased to inform you that your manuscript has been deemed suitable for publication in PLOS ONE. Congratulations! Your manuscript is now with our production department. 

With kind regards,

on behalf of

Dr. Christine E East 

Academic Editor

PLOS ONE